# Effect of Sanding and Plasma Treatment of 3D-Printed Parts on Bonding to Wood with PVAc Adhesive

**DOI:** 10.3390/polym13081211

**Published:** 2021-04-09

**Authors:** Mirko Kariž, Daša Krapež Tomec, Sebastian Dahle, Manja Kitek Kuzman, Milan Šernek, Jure Žigon

**Affiliations:** Biotechnical Faculty, University of Ljubljana, Jamnikarjeva 101, 1000 Ljubljana, Slovenia; dasa.krapez.tomec@bf.uni-lj.si (D.K.T.); sebastian.dahle@bf.uni-lj.si (S.D.); manja.kuzman@bf.uni-lj.si (M.K.K.); milan.sernek@bf.uni-lj.si (M.Š.); jure.zigon@bf.uni-lj.si (J.Ž.)

**Keywords:** 3D printing, adhesive bonding, plasma treatment, beech wood, PLA, ABS

## Abstract

Additive manufacturing is becoming increasingly important for manufacturing end products, not just prototyping. However, the size of 3D-printed products is limited due to available printer sizes and other technological limitations. For example, making furniture from 3D-printed parts and wooden elements requires adequate adhesive joints. Since materials for 3D printing usually do not bond very well with adhesives designed for woodworking, they require special surface preparation to improve adhesion. In this study, fused deposition modelling (FDM) 3D-printed parts made of polylactic acid (PLA), polylactic acid with wood flour additive (Wood-PLA), and acrylonitrile-butadiene-styrene (ABS) polymers were bonded to wood with polyvinyl acetate (PVAc) adhesive. The surfaces of the samples were bonded as either non-treated, sanded, plasma treated, or sanded and plasma treated to evaluate the effect of each surface preparation on the bondability of the 3D-printed surfaces. Different surface preparations affected the bond shear strength in different ways. The plasma treatment significantly reduced water contact angles on all tested printing materials and increased the bond tensile shear strength of the adhesive used. The increase in bond strength was highest for the surfaces that had been both sanded and plasma treated. The highest increase was found for the ABS material (untreated 0.05 MPa; sanded and plasma treated 4.83 MPa) followed by Wood-PLA (from 0.45 MPa to 3.96 MPa) and PLA (from 0.55 MPa to 3.72 MPa). Analysis with a scanning electron microscope showed the smooth surfaces of the 3D-printed parts, which became rougher with sanding with more protruded particles, but plasma treatment partially melted the surface structures on the thermoplastic polymer surfaces.

## 1. Introduction

The applicability of available fused deposition modelling (FDM) 3-dimensional (3D) printers is still limited, particularly in terms of upscaling. Consumers thus typically need to assemble the end-product from multiple smaller parts. Common approaches for furniture pieces like chairs are combinations of complex 3D-printed parts and relatively simple wooden cubic elements. To produce larger products, it is necessary to assemble the final product from smaller 3D-printed parts, combining parts produced by conventional manufacturing methods with less costly materials such as wood. For these approaches, it is important to create a sufficient bond between 3D-printed and wooden parts.

The importance of additive manufacturing (AM) has increased strongly in recent years. AM allows new designs to be created with the rational use of materials and opens up new possibilities for designers, engineers, and do-it-yourself enthusiasts. This is aided by the fact that 3D printers are becoming more accurate, easier to use and more economic, making them more available to a variety of customers and applications. In the past, 3D printing was used for rapid prototyping and focused on the table-top models, where the volume of components was typically around 0.3 to 1 m^3^, printers had slow deposition rates of 15 to 85 cm^3^/h, and prints were priced at 100 to 200 $/kg [1]. An increase in end-user products has been seen in recent years and the technology has developed to such a degree that it is now challenging conventional production techniques for small batch sizes and limited lot production. Even large-scale AM processes with build volumes of 90 m^3^ and more have been demonstrated, including automotive and building applications [1,2].

The related materials and design software are also evolving and becoming more accessible to the general public. Due to increasing environmental awareness, the area of natural-based materials is developing. The use of natural fibers is very convenient, as they are easily available, cheap, environmentally friendly and biodegradable [3]. By using natural fibers or fillers instead of synthetic ones, the gas emissions associated with the production of synthetic components are reduced, which contributes to reducing the current pollution problems [4]. Mazzanti and co-authors [5] reviewed the state of the art on AM using natural fillers and biopolymers, including lignocellulosic compounds, wood material, and other natural fibers, and also provided a brief analysis and possible directions for future research. This emerging field encompasses a growing number of issues to explore, particularly regarding modifications of FDM techniques and the devices that have been used so far [5].

One persisting limitation is the average size of 3D-printed parts. On the one hand, this is due to the limited sizes of commercially available 3D printers. On the other, printing larger objects poses various problems regarding temperature gradients, warping of parts, poor adhesion to the printing bed, increased occurrence of printing errors, and potentially larger amounts of waste material, which is related to higher material costs. To reduce the amount of supporting structures and print parts that are currently too big for 3D commercial printers, it is necessary to slice the models and subsequently bond and assemble the parts using suitable adhesives [6].

Another issue is the printing time of larger objects, which increases with their size. For example, with the used model, printer and slicing software (see details below), a 10% increase in each dimension prolongs the printing time with the same settings by 22%. If the model is increased by 50% in each dimension, the printing time increases by 156%. Therefore, designers are searching for new ways to enable the creation of larger pieces of furniture and home accessory products. One way to reduce the printing time for a complete product is to combine more complex 3D-printed parts and other simple wooden elements produced using conventional methods, such as sawing or milling. As an example, a chair can be efficiently manufactured by only using 3D printing for the more complex connections, whereas the rest consists of simple, long cubic elements such as legs and stretchers, and could thus be made from wood and assembled into the finished product. The assembling requires appropriately designed joints and the utilization of a suitable adhesive to bond 3D-printed and wooden parts. These joints must be strong enough to withstand the loads that occur during their lifetime. The strength of the bonded joints depends on the strengths of the two adhered materials and the strength of the adhesive [7,8,9].

The bond strength of joints between 3D-printed parts and wood strongly depends on the porosity, roughness, layer adhesion, and anisotropy of the 3D-printed parts, which is highly dependent on the 3D printing parameters and printing quality [5,6,8,9]. FDM is characterized by a large number of variables, making it difficult to determine those that most affect structure–property correlations. In addition, the influencing variables may also be material-dependent and interrelated, so further investigation in this direction is essential [4,5,10].

An optimal adhesive must be used to achieve sufficient bond strengths. Good results on bonding 3D-printed parts from acrylonitrile-butadiene-styrene (ABS) to wood were obtained by using two-component polyurethane (PU) adhesive [9], but lower strengths were obtained with hot melt adhesive or one-component PU.

Adhesion to a 3D-printed surface can be increased with proper surface preparation, either during 3D printing or with processes applied afterwards. Kovan and co-authors [6] found that the printing layers’ orientation—i.e., the model’s orientation during 3D printing (edgewise/flatwise)—and layer thickness had significant effects on bond strength.

Proper surface adhesion is required for bonding and coating processes. The coating of AM produced parts is often necessary since the AM process leaves behind individual visual and haptic layers that have to be sanded, filled, sanded again and finally painted [11].

There are a variety of surface methods to control or modify the surface properties of 3D-printed products. Conventional methods include sanding, surface coatings, or pre-treatment with an adhesion promoter. The 3D print can be tailored by reducing the infill within the outer layers in order to increase the effective surface area and produce a porous structure, which supports mechanical interlocking, rather than producing a solid closed layer, which would prevent an adhesive from penetrating the surface and interlocking with the substrate [12]. Other advanced methods include surface alkali hydrolysis treatment, graft polymerization, low-temperature plasma treatment, and various surface chemical reactions [13]. Chemical smoothing can cause a smaller decrease in the tensile strength of the tested specimens, but significantly change the fracture pattern from cohesive substrate failure on non-treated parts to purely adhesive failure on chemically smoothed parts, which is likely due to a difference in surface roughness [14].

Plasma treatments are promising techniques that enable rapid and chemical-free surface modifications [13]. Treatments of polymeric substrates with non-thermal plasma discharges in air increase the surface free energy and improve surface wettability, which is crucial for bonding with liquid water-based adhesives, thus improving the tensile strength of the adhesive bonds [11,14,15,16,17].

Polylactic acid (PLA) polymers were first treated in low-pressure radio-frequency (RF) plasmas [18]. These were particularly successful using ammonia as working gas, which introduces nitrogen-containing groups on the surface that are more hydrophilic than typical oxygen-containing groups generated from air, nitrogen or argon plasmas [19,20]. Other research groups used nitrogen–hydrogen mixtures [21] or even nitrogen and air [22], which resulted in an easier process and lower costs. Further investigations paired RF plasma pretreatments with the immobilization of functional molecules to introduce specific properties to the PLA’s surface [23,24,25]. Another alternative is to introduce the specific groups directly from the plasma, e.g., by using an RF plasma in various fluoron-containing compounds to increase the PLA’s barrier properties [26]. Other plasma techniques used to treat PLA polymer include low-pressure microwave plasmas [27] or low-pressure dielectric barrier discharge (DBD) plasmas [28]. Both techniques were successfully employed to increase the hydrophilicity of PLA through the introduction of polar groups on the surfaces and by increasing the surface roughness, likely due to etching effects. Moreover, the DBD plasma treatment of PLA treatment increased cell adhesion in biotechnical applications [29]. Moraczewski and co-authors [30] compared plasma techniques with laser and chemical modifications, and found that plasma was the most beneficial treatment of PLA, although microwave plasmas required particularly long treatment times of at least 10 min.

In comparison with low-pressure RF and microwave plasmas, atmospheric techniques in general are preferable for many industrial applications including for polymer processing [31]. Sauerbier and co-workers [11] used a DBD operating in synthetic air at atmospheric pressure to treat injection molded PLA specimens. They thus achieved hydrophilization through the generation of oxygen-containing surface groups as well as through an increased surface roughness (i.e., a more developed interfacial area). This further led to increased adhesion of a water-based acrylic dispersion coating, from 1.4 to 2.3 N/mm^2^ [11]. Due to the geometric limitations of DBD plasma devices, remote plasma technologies are used in many industrial sectors. In particular, gliding arc plasma jets can be used independent of substrate geometries for both localized or large-area treatments [32].

Jordá-Vilaplana and co-authors [33] used a commercial gliding arc plasma jet to treat PLA specimens. They observed increased hydrophilicity due to the introduction of polar groups on the surfaces, mainly hydroxyl groups, but also hyperoxide, ether, carboxy acids, and esters. Moreover, they found an increased root mean square (RMS) surface roughness after plasma treatment. Both surface roughness and hydrophilicity were directly correlated with the plasma treatment times.

ABS polymers were etched and functionalized in low pressure RF plasmas for increased platability [34] or for etching purposes [35]. Similarly, treatments of ABS in low pressure RF oxygen plasmas can be used for functionalization, as the treatment leads to the attachment of oxygen containing functional groups, to the removal of nitrile (–C=N) functional groups, to etching, and to an increase in surface roughness [36]. RF plasma can be operated at atmospheric pressure, which has also been demonstrated on ABS [37]. However, gliding arc plasmas and related techniques have better usability for such applications [38], as well as in comparison with surface barrier discharge plasmas [39]. 

The treatment of ABS using an industrial gliding arc plasma jet led to increased hydrophilicity, mainly through the generation of –C=N, –C–O–H, and O–C=O species on the surface, as well as through increased roughness [40]. Further, this treatment was shown to exhibit increased adhesion strengths with epoxy- and PU-based coatings [41], thus enabling compounds (metal–silicone–ABS) to be coated.

Other plasma applications regarding ABS substrates include plasma-deposition of nickel [42], copper [43], increasing the mechanical properties of FDM 3D-printed specimens [15,16,44], ABS nanocomposite synthesis [45], and even the 3D printing of ABS-based plasma probes [46].

In an earlier study [17], it was shown that plasma treatments can enhance bond strengths of mixed material joints and thereby enable new applications for adhesive systems. Specifically, it was shown that low-cost PVAc wood adhesives can be used instead of expensive epoxy adhesives for bonding wood and metal specimens (steel, aluminium) with the same performance.

The aim of the present study was to investigate the bonding of three different 3D-printed materials, namely ABS, PLA and Wood-PLA, with PVAc adhesive to wood. Sanding and plasma treatment were used to enhance surface wettability and adhesion. In order to explain the effects of surface treatments on bonding with wood, the surface roughness was evaluated via confocal laser scanning microscopy, whereas a detailed microstructure of the substrates’ surfaces was analyzed with a scanning electron microscope.

## 2. Materials and Methods

Three commercially available filaments were used: ABS filament Z-ABS (Zortrax, Olsztyn, Poland), PLA (Plastika Trček, Slovenia) and Wood-PLA (wood plastic composite composed of around 40% of wood flour and PLA; Plastika Trček, Slovenia).

Samples from PLA and Wood-PLA were 3D-printed using a Creality CR10-V3 (Creality 3D Technology Co., Ltd., Shenzhen, China) with a direct extruder. The printing layer was set to a thickness of 0.3 mm, the nozzle diameter was 0.4 mm, printing temperature 200 °C, bed temperature 50 °C. Three solid layers were created on the top and bottom, while the infill in the inner layers was 40%. 3D models with the dimensions (50 × 20 × 10) mm and bonding area (20 × 10) mm were modeled in SolidWorks software (SolidWorks Corp., Waltham, MA, USA) and exported to STL format. STL models were sliced and prepared for 3D printing in Cura software (Ultimaker, Utrecht, The Netherlands).

Samples made of ABS were printed using a Zortrax M200 (Zortrax, Olsztyn, Poland) 3D printer, with a nozzle diameter of 0.6 mm, printing temperature of 275 °C, bed temperature of 90 °C, layer thickness of 0.29 mm, three solid layers on the top and bottom, and the infill in the inner layers was 40%. The same STL models as for PLA were sliced and prepared for 3D printing in ABS in Z-Suite software (Zortrax, Olsztyn, Poland).

Samples for determination of bond strength were bonded with the PVAc adhesive Mekol Express (Mitol, Sežana, Slovenia). Adhesive viscosity at 23 °C (ISO 2555-Brookfield RVT, spindle 5/20 rpm) is 7000–10,000 mPa s and density 1.05–1.15 g/cm^3^ (manufacturer data). Wooden adherents were prepared from common beech (*Fagus sylvatica* L.) wood, with the semi-radial orientation of wood fibers, nominal density of 710 ± 18 kg/m^3^ and moisture content of 12 ± 0.5%.

All the analyses explained in the next sections were performed on the surfaces of the following nine different substrates: ABS non-sanded, ABS sanded, ABS plasma treated, PLA non-sanded, PLA sanded, PLA plasma treated, Wood-PLA non-sanded, Wood-PLA sanded, and Wood-PLA plasma treated (Table 1).

### 2.1. Plasma Treatment

The non-thermal atmospheric pressure plasma was used to treat the surface of the 3D-printed samples. The device was manufactured at the University of Ljubljana, Department of Wood Science and Technology, and is presented in Figure 1. The device consists of a computerized numerical control (CNC) positioning system (SainSmart Genmitsu CNC Router 3018 DIY, Vastmind LLC, New Castle, DE, USA) moving the head with attached plasma jet in three directions. Copper electrodes (ROLOT 605, Rothenberger Werkzeuge GmbH, Kelkheim, Germany) are mounted to a 42 mm diameter cylindrical epoxy (Herpelin Epoksi 1000, Amal d.o.o., Ljubljana, Slovenia) nozzle with an 8 mm diameter centred hole as the gas channel. The gas was supplied from an internal compressor (Hailea ACO 208, Guangdong Hailea Group Co., Ltd., Guangdong, China) at a flow rate of approximately 35 L/min. The plasma discharge is generated between the two electrodes within the nozzle using a commercial high voltage module (ZVS_Driver_20A_kit_AC, Voltagezone Electronics e.U., Graz, Austria) operated at an input voltage and current of 20 V and 5 A from a combination of a commercial switch-mode power supply (Joylit S-240-24, Shenzhen Zhaolan Photoelectric Technology Ltd., Shenzhen, China) and a digital power supply (RD DPS5020 BT/USB, Hangzhou Ruideng Technology Co., Ltd., Hangzhou, China). The afterglow of the gliding arc jet extends approximately 2 cm out below the nozzle. During the treatment process, the specimens were placed on the stage with a gap distance between the nozzle outlet and sample surface of 10 mm^2^. The entire surfaces of the samples were treated by the plasma jet scanning in seven lines of 80 mm length offset by 5 mm, thus covering an area of 80 mm × 30 mm, at a moving speed of 60 mm/min. The entire construction details are available in Dahle et al. [47]. The G-code file is provided together with the raw and analyzed data of this publication at Kariz et al. [48].

### 2.2. Scanning Electron Microscopy (SEM)

A scanning electron microscope, FEI Quanta 250 (Thermo Fisher Scientific, Waltham, MA, USA), was used to visually assess the surface microstructure changes between different materials and surface treatments. The samples were sprayed with a gold conductive layer prior to SEM observations. The images of the area on the samples’ surfaces were taken at 100× and 1000× magnification in a high vacuum (1.56 × 10^−2^ Pa), the electron source voltage was 5.0 kV, and the spot size was 3.0 nm. During the capture of each image the time of the beam transition through the sample was 45 μs.

### 2.3. Surface Roughness

Confocal laser scanning microscopy was used to measure changes in the surface morphology by comparing samples before and after plasma treatment. The 3D surface roughness characteristics (average surface roughness parameter *S*_a_ and developed interfacial ratio *S*_dr_) were measured with a confocal laser scanning microscope (CLSM, Olympus, Lext OLS 5000, Tokyo, Japan). An objective lens with 50× magnification was used. Four places on each sample were examined with the area of 20 μm^2^. In order to monitor the changes in surface roughness before and after treatment with plasma, the specimens were analyzed in the same position each time.

The roughness was measured first on sanded surfaces, which were than plasma treated and measured again, so that the same samples were characterized. We analyzed the data for three different materials (ABS, PLA and Wood-PLA), each with two different surfaces (sanded and non-sanded) that resulted in an interfacial area ratio (*S*_dr_).

### 2.4. Water Contact Angle Measurements

The wettability was used as an indicator for the interfacial phenomenon of a liquid contacting a solid surface [49], as represented by the contact angle (CA). One of the most common techniques used for CA determination is the sessile drop method [50], mainly due to its speed, affordability and accuracy. As an indicator, wettability represents the main factor of high importance for achieving good adhesion of coatings and adhesives.

A Theta optical tensiometer (Biolin Scientific Oy, Espoo, Finland, 2016) was used for determination of the CA of a polar liquid with the sessile drop technique. Droplets with a volume of 5 μL of distilled water were dosed. Determination of water CAs followed the established protocol after Nussbaum [51], i.e., determining the angles after a transition time of 5 s after application of the droplets, since it was visually established for the wood specimens that this time interval coincides with the transition from the spreading stage to a constant penetration regime. In contrast to that, no change of CA over a 60 s observation period was notable for the polymer specimens. For each variant, a sample was analyzed using five water droplets. For each such series, average CAs and standard deviations were determined using Microsoft Excel.

The method used is based on Young’s equation:γ_s_ = γ_sl_ + γ_l_ cosθ(1)
where γ is the surface tension or interfacial energy (mJ m^−2^) of the solid–vapor (s), the solid–liquid (sl) and the liquid–vapor (l) interfaces, respectively. In principle, Young’s equation assumes that the entire system is at thermodynamic equilibrium and that the solid surface is chemically homogeneous, flat and not influenced by chemical interactions or adsorption of the liquid to the surface [52].

### 2.5. Adhesive Bonding

Before bonding, 3D-printed samples were divided into four groups for different surface preparations: one group kept the surface as it came out of the 3D printer, one was plasma treated, one group was manually lightly sanded with 150 grit sandpaper to ensure active and flat bonding surfaces, and one group’s surfaces were lightly sanded and plasma treated.

PVAc adhesive was used, since this is a common adhesive in the production of wooden furniture, relatively simple to use, does not contain harmful substances and is also environmentally friendly. The adhesive application rate was 180 g/m^2^, pressing time 1 h with 6 MPa pressure in a laboratory press at room temperature. After bonding, the samples were left in a conditioning chamber at 20 °C and 65% of air relative humidity for seven days before testing on a Zwick Z005 universal testing machine (ZwickRoell GmbH, Ulm, Germany).

To determine the bond strength of the specimens, tensile-shear tests were performed according to a modified testing standard SIST EN 205 [53] with a test speed of 50 mm/min. Ten specimens per series were tested and results analyzed with Microsoft Excel and Statgraphics software.

## 3. Results and Discussion

The results are presented and discussed, starting with the basic observations of the used treatments’ effects on microstructure and roughness, as these directly affect the later measurements. Based on these results, the water CAs of the optimized treatments are discussed in relation to the surfaces morphology and chemistry. Finally, the adhesive bond strength of the joints are investigated and the underlying mechanisms are discussed based on the overall findings.

### 3.1. Surface Microstructures

The non-sanded surfaces appear smooth, exhibiting strands of filament for each printing line (Figure 2). The surface is smooth, since the material was deposited by extruding molten polymers that spread evenly. There are smaller particles present, probably remaining from previous printing processes, originating from the printing nozzle.

The sanding removed only the top of the printed strands, and especially on PLA the rough surface shows the outer layers of the strands removed by sanding with sanding paper, whereas the edges of each strand (shown as valleys) were still smooth (Figure 3), since not enough material was removed to reach this depth. There are clearly seen scratches on all sanded surfaces and particles that protrude. These are deformed particles, which were not removed completely.

There is not much visual difference seen on the non-sanded surfaces before and after plasma treatment. On sanded surfaces, the particles that protrude from the surface from the remains of the sanding are smaller and have more melted edges (Figure 4 and Figure 5)**.** This agrees with the existing literature, which shows that plasma treatment used to etch PLA surfaces and extended plasma exposure times lead to the formation of oligomers and desorption of volatile products from the PLA surface (etching) [28].

The surface of the Wood-PLA samples is the least smooth (Figure 6 and Figure 7), showing a high porosity and open voids, which are typical for wood plastic composites [54,55]. The voids on the surfaces found using SEM were mostly caused by the moisture contained in the filler material [10]; wood fibers have a hydrophilic nature and attract water, which changes to vapor state during extrusion and thus forms these defects in the FLM 3D-printed sample. It is thus very important to store Wood-PLA filaments in closed containers to prevent moisture intake or dry them before 3D printing. Some of the porosity originates from filament production due to the lack of a mixing state during the fabrication of the wood-based filament, and due to the lack of melting and blending pressure in the FDM extrusion process [54,55]. Voids and porosity contribute greatly to the reduction in mechanical stiffness and strength [56], since fractures tend to initiate within them. 3D-printed biocomposites could have a microstructure with relatively high porosity (even around 20%) that conjointly leads to damage mechanisms, but also to high and faster water absorption and swelling [54]. Moreover, this microstructure is the likely origin for the higher water contact angles and lower average bond strengths observed in the Wood-PLA specimens, despite the presence of wood particles that were expected to be more hydrophilic and bondable than the surrounding PLA polymer. Furthermore, the reduced surface roughness (Sa) after plasma treatments seem to be related to a partial melting of the polymer surface, and consequently a partial closure of the observed voids. The degree to which the increased adhesive bond strength can be attributed to this observation, however, remains unclear.

### 3.2. Surface Roughness

Surface roughness (Table 1) is an important factor for adhesive bonding. On the one hand, the wetting of the surface, the interfacial area, and the mechanical interlocking with the adhesive are all increased on rough surfaces due to the larger specific surface area. In ISO 25178-2 [57] the surface roughness parameter Sa is defined as the arithmetic mean of the absolute of the ordinate values within a defined area, which is used generally to evaluate surface roughness. On the other hand, too high surface roughness inhibits contact between the joined surfaces and thereby impairs the adhesion strength.

Depending on the sanding grit used, sanding decreases the surface roughness by removing the highest peaks on the surface or increases the surface roughness due to the formation of small scratches on the surface. The plasma treatment is known to etch the PLA surface [28], but this effect depends on the type of plasma discharge used. Moreover, on substrates with a low melting point and glass transition temperature, the increase in temperature on the treated surface during the exposure to the plasma discharge could heat the surface sufficiently to locally melt the material and thereby decrease the surface roughness.

### 3.3. Water Contact Angles

Figure 8 shows the CAs of all samples systems as printed, sanded, plasma treated, as well as both sanded and plasma treated. All the tested materials followed similar trends in that sanding increased the water contact angle (CA), while plasma treatments decreased it. ABS typically shows a CA of approximately 60° [58], whereas PLA is a relatively hydrophobic polymer with a static water contact angle in the range of 75–85° [13]. These are well represented by the initial WCA measured on as-printed specimens, whereas the additional roughness from sanding increased the CA through the decreased effective contact area [59], assuming a Wenzel mode of wetting [60,61]. The highest CA of 125° was measured on the sanded ABS samples. The lowest CA of 16° was measured on sanded and plasma treated ABS. On pure ABS and PLA polymers, combinations of sanding and plasma treatment led to lower CA than seen with plasma treatments without sanding, whereas for the plasma treated Wood-PLA specimens the CA was the same with or without sanding. The effects of both sanding and plasma treatments were less pronounced on the Wood-PLA. Regarding sanding, this is likely due to the mechanical reinforcement by the wood particles. In contrast, the plasma treatment of Wood-PLA is likely to be influenced by the chemistry of the wood particles, which are expected to dominate the interfacial interaction and thus determine the CA after plasma treatment to a large degree, regardless of the previous sanding.

The effect of increasing surface hydrophilicity with plasma treatment is well known and expected. Sauerbier and co-authors [11] reported that water contact angles on a PLA-based wood-polymer composite (WPC) with 60% wood content decreased significantly after plasma treatment. A similar effect of plasmas is reported on ABS [15]. This effect depends on the type of plasma and on exposure times. Nitrogen discharges plasmas tend to improve wettability even more than air and argon plasmas [28]. In comparison to a commercial plasma device, the treatment speeds in this study are significantly lower due to the lower electrical power throughput and hence the lower treatment intensities. However, the similarity of the plasma discharges and the well-reproduced contact angles indicate a similar chemistry, which includes the addition of mainly hydroxyl, hyperoxide and ether groups to the PLA surfaces [33], as well as hydroxyl, carbonyl, carboxyl, and nitrile groups on ABS surfaces [40].

The PVAc adhesive used in this research has a strong hydrophilic character due to hydroxy and carboxy groups appended to the polymer backbone, as well as due to the addition of polyvinyl alcohol to form a water emulsion [62]. It is thus expected to achieve better wettability on hydrophilic surfaces, which is a prerequisite for the adhesion between any substrate and the adhesive.

Wood is composed of cellulose, hemicellulose, and lignin, as well as extractives and a small amount of inorganic components [63], which include numerous hydroxy groups and thus has a strong polar, i.e., hydrophilic, character [64]. Therefore, the wettability of wood with PVAc adhesive is good, despite the fact that liquid wetting on complex substrates like wood is influenced by several factors like surface thermodynamics, roughness, porosity, heterogeneity, bulk sorption, liquid viscosity, reorientation of functional groups at the wood–liquid interface, and contamination of the wetting liquid by wood extractives [65].

Previous studies reported that the water contact angles values for PLA varied from 60° to 85° [66]. A similar result was found in research from Ayrilmis and co-authors [3], where the contact angle values of the PLA and ABS specimens were found to be 67.8° and 79.6°, and the contact angles of wood/PLA specimens with a wood flour content of up to 30 wt% were found to be lower than 90°. A low contact angle is very important (particularly below 90°) in achieving a strong adhesion bond between coatings and the substrate surface. Liquid coatings such as paints, varnishes and adhesives should wet the substrate surface sufficiently for physical adhesion [3]. The surface properties of parts affect their wettability, and the surface quality is affected by process parameters of FDM 3D-printed parts. A high surface free energy of the adherent and a low surface energy of the adhesive are desirable, as these conditions promote wetting and spreading of the adhesive [67].

### 3.4. Bond Tensile-Shear Strength

The bond shear strength of samples directly from 3D printing (non-sanded, non-plasma treated) was low, ranging from 0.45 ± 0.3 MPa for Wood-PLA, 0.55 ± 0.21 MPa for PLA, and 0.05 ± 0.16 MPa for ABS (Figure 9). Sanding the 3D-printed surface increased bond strengths for all materials; the highest bond shear strength was found for Wood-PLA samples with 2.98 ± 0.47 MPa, followed by PLA 2.69 ± 0.63 MPa and the lowest for ABS samples 1.8 ± 1.17 MPa. The highest strength for Wood-PLA was expected, since we used a commercial adhesive intended for bonding of porous materials like wood. PVAc adhesives bond to the hydrophilic surfaces of wooden particles and probably also penetrate into them, thus mechanically interlocking and increasing the adhesion strength. Sanding exposes the wooden particles, and thus better adhesion was expected with the sanded surfaces. The mechanical properties of the Wood-PLA are the lowest [68], but since failure never occurred in the adhered material this did not affect the measured strength. PLA and especially ABS exhibit hydrophobic surfaces, to which water-based adhesives do not sufficiently adhere.

After the plasma treatment, the highest strength was determined for ABS samples with 4.83 ± 1.35 MPa, followed by Wood-PLA at 3.96 ± 0.36 MPa, and PLA at 3.72 ± 0.67 MPa. The plasma treatment increased the bond strength for selected 3D-printed materials, but in different ratios. The highest increase from basic 3D-printed samples was found for ABS (by 95.5 times), for PLA it increased by 8.8 times and for Wood-PLA by 6.7 times. Due to the adhesive transparency of the cured adhesive, it was difficult to determine whether the failure occurred in the adhesive or in the contact to adherent surface, but failure never occurred in the printed material.

In addition to the highest increase in bond shear strength, there was also the highest change in contact angles on ABS samples after plasma treatment. Comparable positive effects of plasma treatment on adhesion and bond shear strength on different printed surfaces and with different adhesives were reported before [11,14,17]. Overall, the measured adhesive bond strengths are very well in line with the contact angles.

## 4. Conclusions

Surface preparation changed the properties of the 3D-printed samples’ surfaces and affected bonding. Sanding increased the contact angles of deposited water droplets, but plasma treatment made the surfaces more hydrophilic. The decrease in contact angle after plasma treatment was the highest for sanded ABS (86%), high also for sanded PLA (55%) and Wood-PLA (49%), notable for non-sanded ABS (40%), non-sanded Wood-PLA (28%) and lowest for non-sanded PLA (16%). The results show that plasma treatment significantly reduces the contact angle on the surfaces of the tested printing materials, and thus increases the wettability.

Different surface preparation methods affected the bond shear strength. The bond shear strength of samples directly from 3D printing (non-sanded, non-plasma treated) was low (from 0.45 MPa for Wood-PLA, 0.55 MPa for PLA and 0.05 MPa for ABS). Preparation with sanding increased the strength of all tested materials. The highest bond strengths were found on sanded and plasma treated surfaces of ABS (4.83 ± 1.35 MPa), followed by Wood-PLA (3.96 ± 0.36 MPa) and PLA (3.72 ± 0.67 MPa). 

The results were as expected, since the PVAc adhesive used adheres to hydrophilic surfaces and plasma treatment increased the hydrophilicity of the surfaces, while sanding also roughened the surfaces and enabled more mechanical interlocking of adhesive. The increase in bond strength was highest for the ABS samples, which also exhibited the highest change in WCA. The positive effect of plasma treatment on adhesion and bond shear strength for different printed surfaces and with different adhesives was reported before [11,14,17], particularly highlighting the effect of polar chemical groups on the surfaces produced by the plasma treatments [33,40].

Visual assessment of material surfaces as photographed with SEM showed that sanding removed the outer layer of the 3D printing strands and the surface looked rougher. However, the surface roughness measurements showed that sanding decreased the surface roughness for ABS and PLA but increased it for Wood-PLA. Plasma treatment melted some of the edges of protruding particles and thereby decreased the apparent surface roughness. This could also have contributed to improving surface contact with the adhesive formulation, and to increased adhesive bond strength.

The results show that suitable surface preparation is essential in achieving sufficient bond strength. Sanding significantly increased the bond strength, but the combination of sanding and plasma treatment gave the best results.

## Figures and Tables

**Figure 1 polymers-13-01211-f001:**
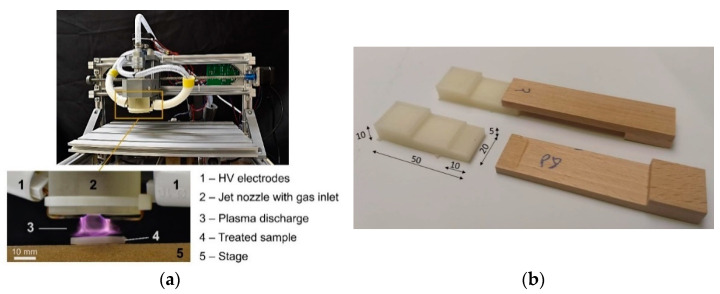
Plasma device used for the surface treatment of 3D-printed samples on a CNC positioning system (top left image) with a gliding arc plasma jet (**a**), and sample for testing the bond shear strength (**b**).

**Figure 2 polymers-13-01211-f002:**
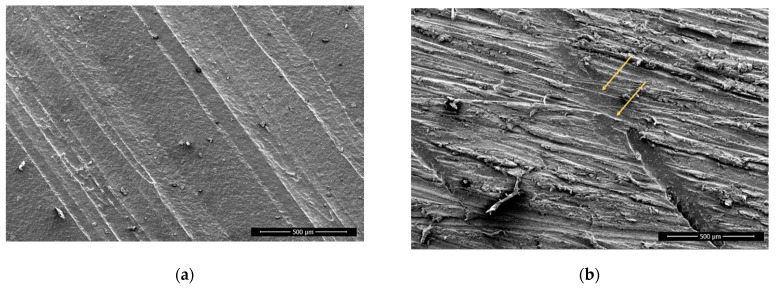
ABS, non-sanded, non-treated (**a**) and sanded (**b**), 100× magnification; arrows pointing to the sanding traces.

**Figure 3 polymers-13-01211-f003:**
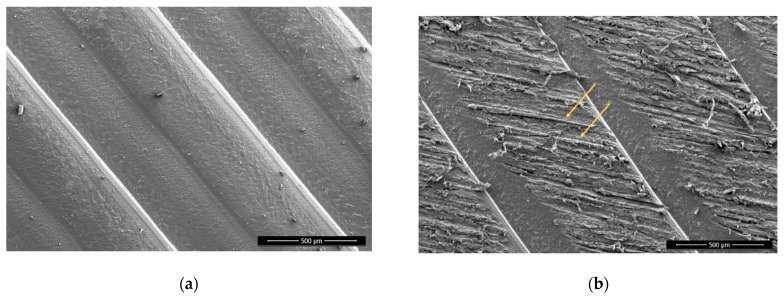
PLA, non-sanded, non-treated (**a**) and, sanded, non-treated, with visible sanding traces; (**b**) 100× magnification.

**Figure 4 polymers-13-01211-f004:**
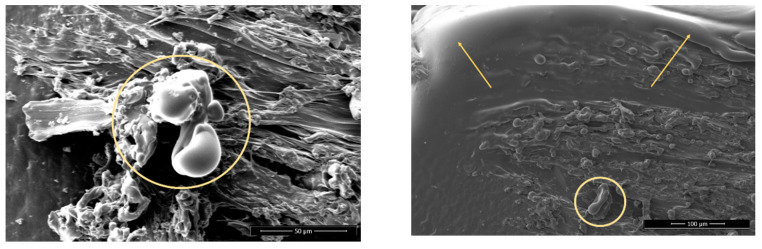
PLA, sanded, plasma treated, 1000× magnification. Melted particles are encircled, arrows are pointing the melted material edges with rounded appearance.

**Figure 5 polymers-13-01211-f005:**
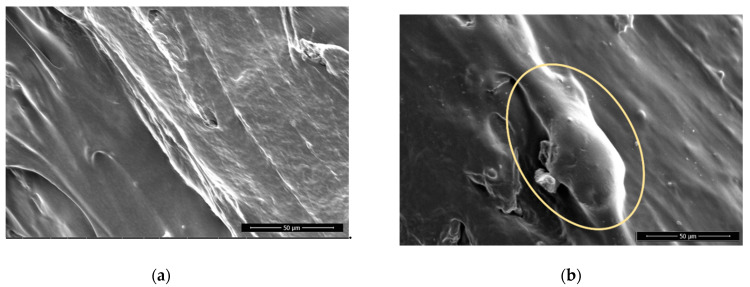
Wood-PLA, non-sanded, non-treated (**a**) and plasma treated (**b**), 1000× magnification. Melted materials are encircled.

**Figure 6 polymers-13-01211-f006:**
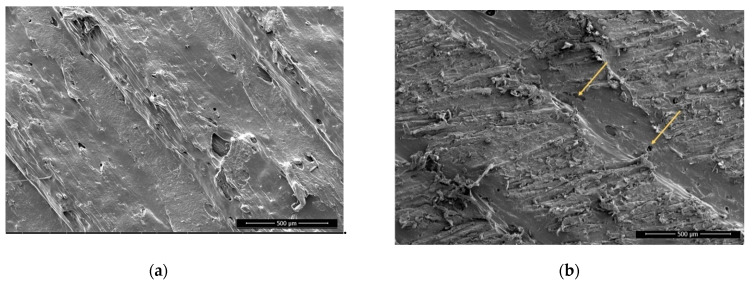
Wood-PLA, non-sanded, non-treated (**a**) and sanded (**b**) 100× magnification. Arrows showing high porosity and open voids in surface.

**Figure 7 polymers-13-01211-f007:**
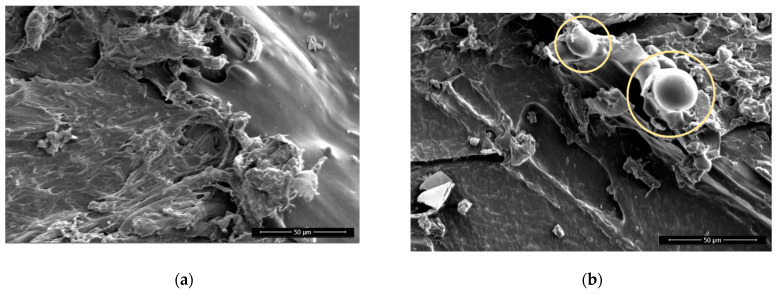
Wood-PLA, sanded, non-treated (**a**) and plasma treated with visible melted particles (**b**), 1000× magnification. Melted particles are encircled.

**Figure 8 polymers-13-01211-f008:**
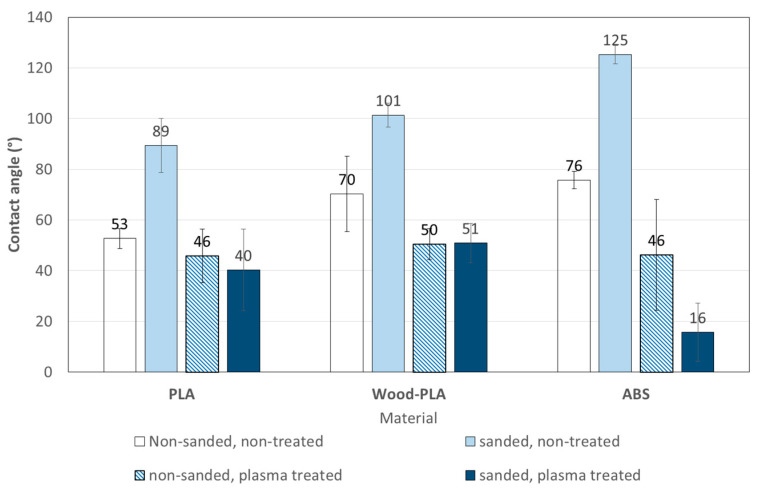
Averaged contact angles of water droplets, measured 5 s after the sessile drop.

**Figure 9 polymers-13-01211-f009:**
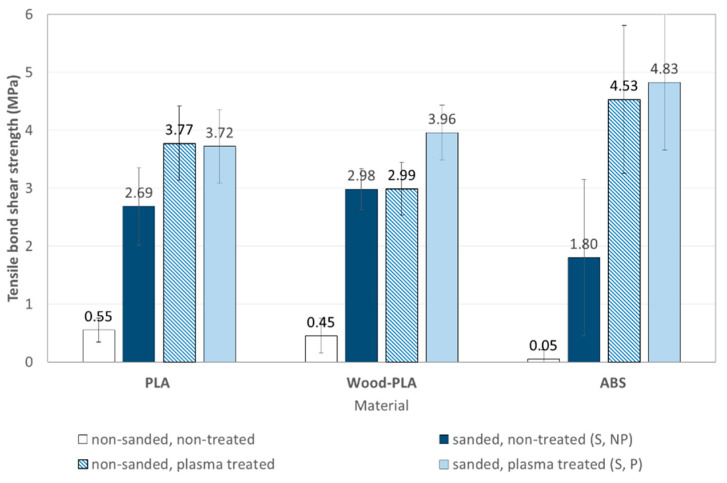
Bond tensile-shear strength of sanded samples from PLA, Wood-PLA and ABS, non-treated and treated with plasma.

**Table 1 polymers-13-01211-t001:** Average surface roughness parameters *S*_a_ and *S*_dr_ measured on samples from acrylonitrile-butadiene-styrene (ABS), polylactic acid (PLA) and Wood-PLA, non-sanded (NS) and sanded (S) surface, non-treated (NP) and treated with plasma (P).

	NP	P			*S*_dr_ NP[%]-*S*_dr_ P[%]	
	*S*_a_ [µm]	*S*_a_ [µm]	avg ∆	STD	avg ∆	STD
PLA, NS	14.85	13.87	0.99	0.48	45.83	29.95
PLA, S	14.66	14.48	0.18	1.24	−1.37	34.72
Wood-PLA, NS	25.03	26.37	−1.34	1.33	−48.37	51.95
Wood-PLA, S	19.03	19.95	−0.92	1.00	−34.94	22.69
ABS, NS	26.23	28.15	−1.92	3.32	−42.28	32.43
ABS, S	17.10	16.89	0.21	2.35	−37.32	23.88

## Data Availability

The data presented in this study are available on request from the corresponding author.

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
