# Peer review of "Effect of Sanding and Plasma Treatment of 3D-Printed Parts on Bonding to Wood with PVAc Adhesive"

_polymers, 2021, doi:10.3390/polym13081211_

Round 1

Reviewer 1 Report

The aim of the present work was thus to investigate the bonding of three different 3D-printed materials (ABS, PLA and Wood-PLA) with PVAc adhesive to wood. Sanding and plasma treatment were used to enhance surface wettability and adhesion. Some additions in the text are necessary. 

COMMENTS

Key words

  • It will be better to use the name of wood specie used for the investigation – “beech wood” than only “wood”.

Introduction

  • The state of the art correctly has been presented.

Materials and Methods

  • Properties of the commercial PVAc adhesive (e.g. viscosity, density) should be given.
  • Authors wrote „…beech (Fagus sylvatica L.) wood, with the semi-radial orientation of wood fibres, nominal density of 710 kg/m3 and moisture content of 12%...” It will be better to give the range of density and moisture content e.g. 710±…kg/m3, and 12±…%.
  • Authors should explain why contact angles were analysed 0 - 5 seconds after the droplets came into contact with the surfaces ?

Results and Discussion

  • The investigation results were correctly and clearly presented. Authors should add information about statistical estimation method.

Conclusions

  • Conclusions corresponds with the aim of the work. Authors should change the chapter number (4 - not 5).

References

  • Authors correctly selected and cited 66 literature sources.

Based on the above considerations my recommendation is: accept after minor additions.

Author Response

We thank the reviewers for remarks, comments, questions and suggestions on our manuscript. Answers to reviewer 1 comments are in attached document.

Reviewer 2 Report

The manuscript is focused on an important and innovative research topic, namely investigation of the bonding of three different 3D-printed materials, namely acrylonitrile-butadiene-styrene (ABS), polylactic acid (PLA), and polylactic acid with wood flour additive (Wood-PLA), bonded to wood with PVA adhesive. In this respect, the topic of the manuscript is relevant and appropriate to the aims and objectives of the Special Issue "3D Printing in Wood Science" of Polymers Journal. In the recent years, the additive manufacturing has gained a significant industrial interest, including application in furniture industry.

The abstract of the manuscript (lines 9 to 27) and the keywords (line 28) correspond to the title, aims and objectives of the manuscript. I would recommend to replace the abbreviations PLA and ABS in the keywords with the full terms.

In line 47, the phrase “However, recent years have seen an increase in…” should be revised, e.g. “An increase in end-user products has been….”.

In line 54, “natural based” should be hyphenated, i.e. “natural-based”.

In lines 73-75, the sentence “For example, if the model increases 10% in each dimension, the printing time with the same settings increases 22%; if the model increases 50% in each dimension, the printing time increases 156%.” should be revised, e.g. “For example, if the model is increased by 10% in each dimension, the printing time with the same settings increases by 22%; if the model is increased by 50% in each dimension, the printing time increases by 156%.”. This statement is probably true and based on authors’ previous experience, but I recommend to add relevant references.

In lines 87-89, the statement “The bond strength of joints between 3D-printed parts and wood strongly depends on the porosity, roughness, layer adhesion, and anisotropy of the 3D-printed parts, which is highly dependent on the 3D printing parameters and printing quality.” should also be followed by appropriate reference(s).

In line 177, please revise “strengths for mixed material” to “strengths of mixed material”.

In line 181, please either delete the word “thus” or place it at the beginning of the sentence.

In line 208, please use italics for the botanical name of the beech, i.e. Fagus sylvatica L.

In line 213 (page 5) you mention Table 1, which, however, is located on page 11. This somehow distracts the readers. I would recommend not to include Table 1 in line 213.

In line 302, the specified testing standard SIST EN 205 should also be added in the references.

In general, the Materials and Methods section is presented extremely well and contains all necessary information.

In line 312, please revise “Fig. 2” to “Figure 2”. The same remark applies to Figures 2 – 7. These figures should be properly addressed in the text of the manuscript. I would recommend to include a sentence or a short paragraph from the very beginning of Results and Discussion.

In line 389, I believe the authors made a typo mistake, i.e. Figure 2 instead of Figure 8. Please, revise.

In line 468, please revise “Fig 9” to “Figure 9”, and address it in the text of the manuscript.

Overall, the Results and Discussion part is very well developed and properly discussed.

The Conclusions are consistent with the results and reflect the main findings of the study.

The references cited are appropriate and correspond to the topic of the manuscript. The inclusion of additional references at certain points in the manuscript, as stated above, will greatly contribute to the quality of the presented manuscript.

Author Response

We thank the reviewers for remarks, comments, questions and suggestions on our manuscript. Answers to reviewer 2 comments are in attached document.
